# Learning to predict future locations with internally generated theta sequences

**Eloy Parra-Barrero** [1,2], **Sen Cheng** [1,2]*

1 Institute for Neural Computation, Faculty of Computer Science, Ruhr University Bochum, Bochum, Germany, 2 International Graduate School of Neuroscience, Ruhr University Bochum, Bochum, Germany

* sen.cheng@rub.de

## Abstract

Representing past, present and future locations is key for spatial navigation. Indeed, within each cycle of the theta oscillation, the population of hippocampal place cells appears to represent trajectories starting behind the current position of the animal and sweeping ahead of it. In particular, we reported recently that the position represented by CA1 place cells at a given theta phase corresponds to the location where animals were or will be located at a fixed time interval into the past or future assuming the animal ran at its typical, not the current, speed through that part of the environment. This coding scheme leads to longer theta trajectories, larger place fields and shallower phase precession in areas where animals typically run faster. Here we present a mechanistic computational model that accounts for these experimental observations. The model consists of a continuous attractor network with short-term synaptic facilitation and depression that internally generates theta sequences that advance at a fixed pace. Spatial locations are then mapped onto the active units via modified Hebbian plasticity. As a result, neighboring units become associated with spatial locations further apart where animals run faster, reproducing our earlier experimental results. The model also accounts for the higher density of place fields generally observed where animals slow down, such as around rewards. Furthermore, our modeling results reveal that an artifact of the decoding analysis might be partly responsible for the observation that theta trajectories start behind the animal's current position. Overall, our results shed light on how the hippocampal code might arise from the interplay between behavior, sensory input and predefined network dynamics.

**Data Availability Statement:** All code written in support of this publication is publicly available at https://github.com/EloyPB/theta_sequence_models.

## Author summary

To navigate in space we need to know where we are, but also where we are going and, possibly, where we are coming from. In mammals, including humans, this might rely on the hippocampal theta phase code, where in each cycle of the theta oscillation, spatial representations appear to start behind the animal's location and then sweep forward. Previously, we showed that these sweeps extend to the locations that were or will be reached at fixed time intervals in the past or future, but assuming the animal ran at its typical speed through each portion of the environment. Here, we present a computational model that

**Funding:** This work was supported by grants from the German Research Foundation (Deutsche Forschungsgemeinschaft, DFG), project number 419037518 – FOR 2812, P2 (S.C.) — and project number 122679504 – SFB 874, B2 (S.C.). The latter payed for the salary of E.P.B. The funders had no role in study design, data collection and analysis, decision to publish, or preparation of the manuscript.

**Competing interests:** The authors have declared that no competing interests exist.

can account for these effects, as well as for the over-representation of reward zones in the hippocampal code. The model uses preconfigured neural sequences in the hippocampus to learn sequences of spatial inputs, a mechanism which is supported by experimental findings. Similar mechanisms have been proposed to underlie the encoding of episodic memories. Our work might therefore help reconcile the prominence of spatial representations in the hippocampus with its well-known function in episodic memory.

## Introduction

Hippocampal place cells elevate their firing rate within circumscribed regions of the environment known as the cells' place fields [1, 2]. In addition to this firing rate code, place cells display a form of temporal coding. As an animal crosses a cell's place field, the cell fires at progressively earlier phases of the theta oscillation in a phenomenon known as theta phase precession [3]. In a population of such phase precessing cells, the first cells to fire within each theta cycle have place fields centered behind the current position of the animal, followed by cells with place fields centered progressively ahead [4–6]. Decoding the positions represented by the population activity during these theta sequences reveals trajectories that start behind the current position of the animal and sweep ahead [7–14]. Previously, we found that theta sweeps in a population of CA1 place cells vary systematically across the environment [15]. In particular, place cell activity was best accounted for by a behavior-dependent sweep model according to which theta sweeps start at the position the animal reached a fixed time interval into the past, and extend to the position the animal would reach a fixed time interval into the future, assuming the animal ran at the speed that is typical for the current location of the animal. As a result, theta sweeps cover more ground in areas characterized by faster running. At the single cell level, this is expressed as larger place fields with shallower phase precession in those areas (Fig 1). Place cell properties, however, do not change with instantaneous running speed, i.e., they are not affected by the animal running slower or faster than usual on a given trial.

This theta phase code has been proposed to play a role in a wide array of cognitive functions, such as working memory [16], sequence learning [4, 16, 17], prediction [12, 18], and planning [19–22]. However, its underlying neural mechanisms remain a matter of discussion [23]. Some have argued that theta sweeps are the result of a population of place cells that phase precess independently from one another [24, 25]. One way in which this could occur is through the oscillatory interference pattern created within cells receiving two oscillatory inputs of slightly different frequencies [3, 26]. Another way of generating phase precession at the single-cell level involves the competition between an oscillatory inhibitory input, and an excitatory input that ramps up as the animal crosses the cell's place field [27–32]. As excitation increases, inhibition is overcome at progressively earlier phases of the theta cycle, resulting in phase precession. A potential issue with this kind of model is that place cell activity is always tied to external inputs, and therefore activity at the late phases of theta cannot truly be said to predict upcoming positions as it is sometimes believed to be the case.

A different kind of model focuses on network connectivity and proposes that phase precession results from theta sequences generated by the propagation of activity in the network through synaptic connections between cells (Fig 2A) [33–36]. In this kind of model, it is often assumed that place cells are first activated sequentially by changing sensory and idiothetic inputs and later linked via lateral synaptic connections that are learned in the first trials in a novel environment (Fig 2B, top). An intriguing alternative is that the sequences are pre-configured in the network, and what is learned is primarily the association between the external

**Average Temporal Sweep**

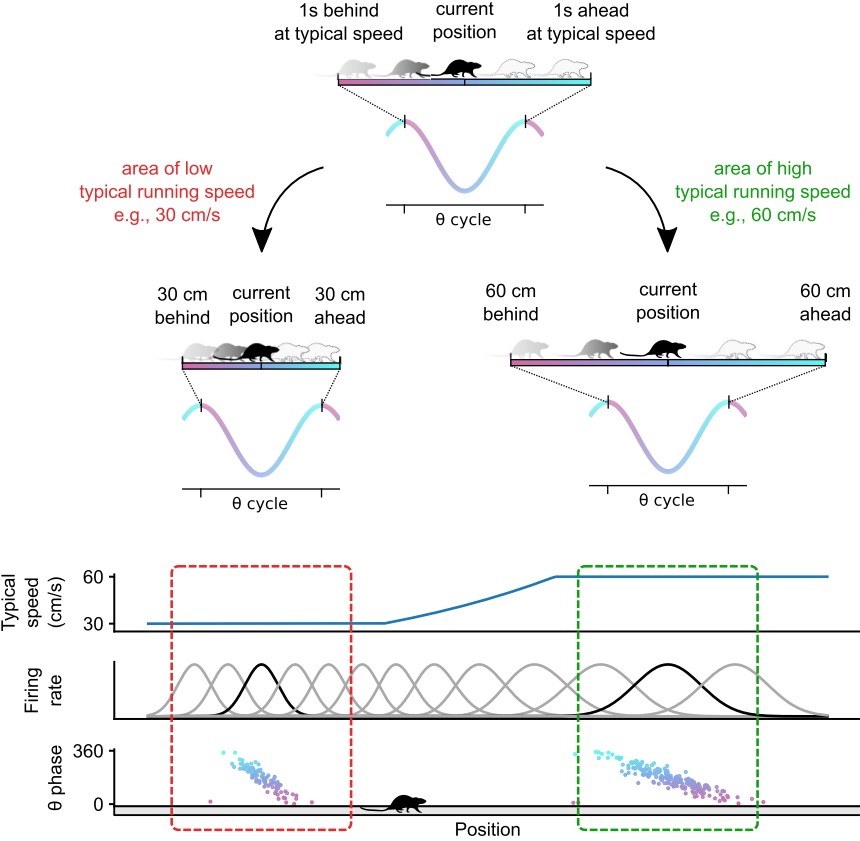

**Fig 1. Illustration of behavior-dependent sweeps reported in [15].** Within each theta cycle, the trajectory represented by the place cell population starts at the position where the animal was located at some point in the past (e.g., 1 s ago), and sweeps to the position where the animal will be located at at some point in the future (e.g., in 1 s) assuming the animal ran at its typical speed at each position (top). As a result, theta sweeps are shorter where animals typically run slower (middle left) and longer where they typically run faster (middle right). At the single cell level, this effect manifests as smaller place fields with steeper phase precession where animals run slower (bottom left) and larger place fields with shallower phase precession where animals run faster (bottom right).

inputs and the cells in the sequence (Fig 2B, bottom) [37–39]. Some support for the latter alternative comes from the observation that the hippocampus can internally generate sequences of time cells [40, 41]. Most notably, however, the idea is supported by the observation of preplay: cells that develop a sequence of place fields on a novel linear track already fired in a similar order during sharp wave ripples before the animal ever experienced the track [42–46]. Here we develop a mechanistic computational model that operates based on this principle, mapping out space with pre-existing internally generated sequences. We show that the model can account for behavior-dependent sweeps and offer an explanation for the higher density of place fields observed around rewards and other areas where animals slow down [47–61].

## Results

### Elastic mapping of space by internally generated sequences

The principle underlying the proposed model is sketched in Fig 2C. Activity propagating at a fixed pace in a 1D neural space is used to map positions along a linear track. If the animal

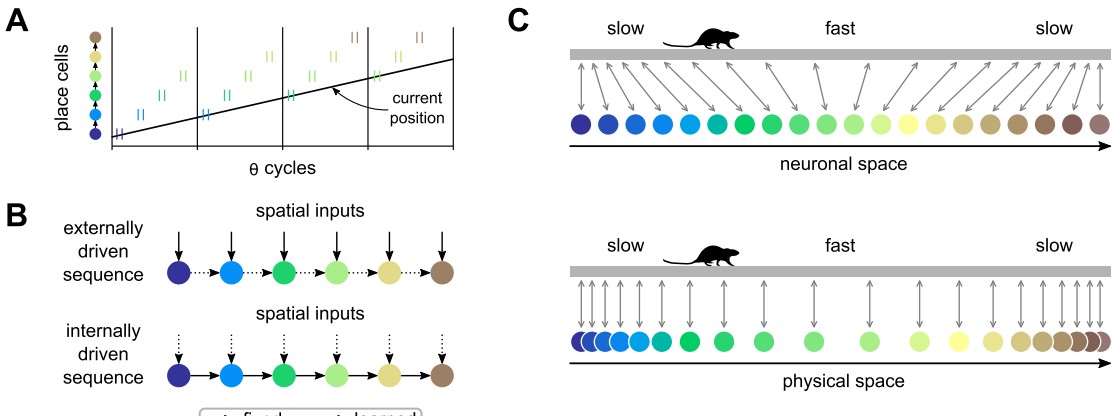

**Fig 2. Network connectivity models of phase precession and proposed mode of operation.** A: In the network connectivity model of phase precession, each theta cycle starts with spikes (short vertical lines) of place cells that code for the current position of the animal, that is, cells that receive strong external inputs at that location. Activity then propagates forward through synaptic connections to cells coding for positions progressively ahead. As a result, an individual cell fires at progressively earlier positions within the theta sequences as the animal approaches the location encoded by a cell, generating phase precession. B: Two variants of network connectivity models. Either sequences of place fields are initially driven by external spatial inputs and the connections between cells are learned afterwards, or the connections between cells are present from the start and the association with spatial inputs is learned afterwards. C: Principle underlying our model of behavior-dependent sweeps. A rat runs on a linear track with a characteristic slow-fast-slow speed profile. Top: If activity in an internally driven sequence propagates at a fixed pace in neuronal space, neighbouring units become associated with positions closer to each other where the animal runs slowly and further apart where the animal runs faster. Bottom: Same as above, but displaying each neuron at the position it becomes associated with. The neuronal space could be seen as elastic, becoming stretched out in areas of high running speed, and compressed in areas of low running speed.

typically runs at different speeds through different parts of the track, the mapping between neuronal and physical space will be uneven. In particular, if activity initially propagates at a pace of $u_0$ neurons per second, and the animal runs on average at $\bar{v}$ cm per second through some portion of the environment, then $\frac{u_0}{\bar{v}}$ neurons per cm will become associated with that portion. In other words, neighbouring neurons will become associated with positions further apart the faster the animal runs through those positions on average. If the network additionally generates theta sequences that have a fixed length in neuronal space, these sequences will appear longer in physical space where the animal ran faster on average. As a result, place fields will be wider and theta phase precession will be shallower in those areas, consistent with our earlier experimental results [15].

Our network model contains an input layer of cells that are weakly spatially tuned and a layer of place cells (Fig 3A). The activity of input cells is modeled as frozen noise of different spatial frequencies, which could be seen as corresponding to a mixture of entorhinal cortex inputs, e.g., from border cells [62], grid cells of different spatial scales [63] or object cells [64]. The place cell layer is adapted from a model of CA3 developed by Romani and colleagues [36, 65]. It is a recurrent network with short-term synaptic facilitation and depression that is pre-configured to generate sequential activity that advances both within and across theta cycles (Fig 3D, bottom left). Within theta cycles, an activity bump forms and propagates forward in the network due to slightly asymmetric recurrent connections (Fig 3B) and short-term synaptic depression. Between theta cycles, activity in the network is shut down by bouts of inhibition (Fig 3C, gray curve). These could correspond to decreases in the input from GABAergic cells of the medial septum targeting hippocampal interneurons [66]. When inhibition recedes, the interplay between a slower short-term synaptic facilitation and a faster short-term synaptic depression ensures that a new activity bump is formed slightly ahead of where the last one

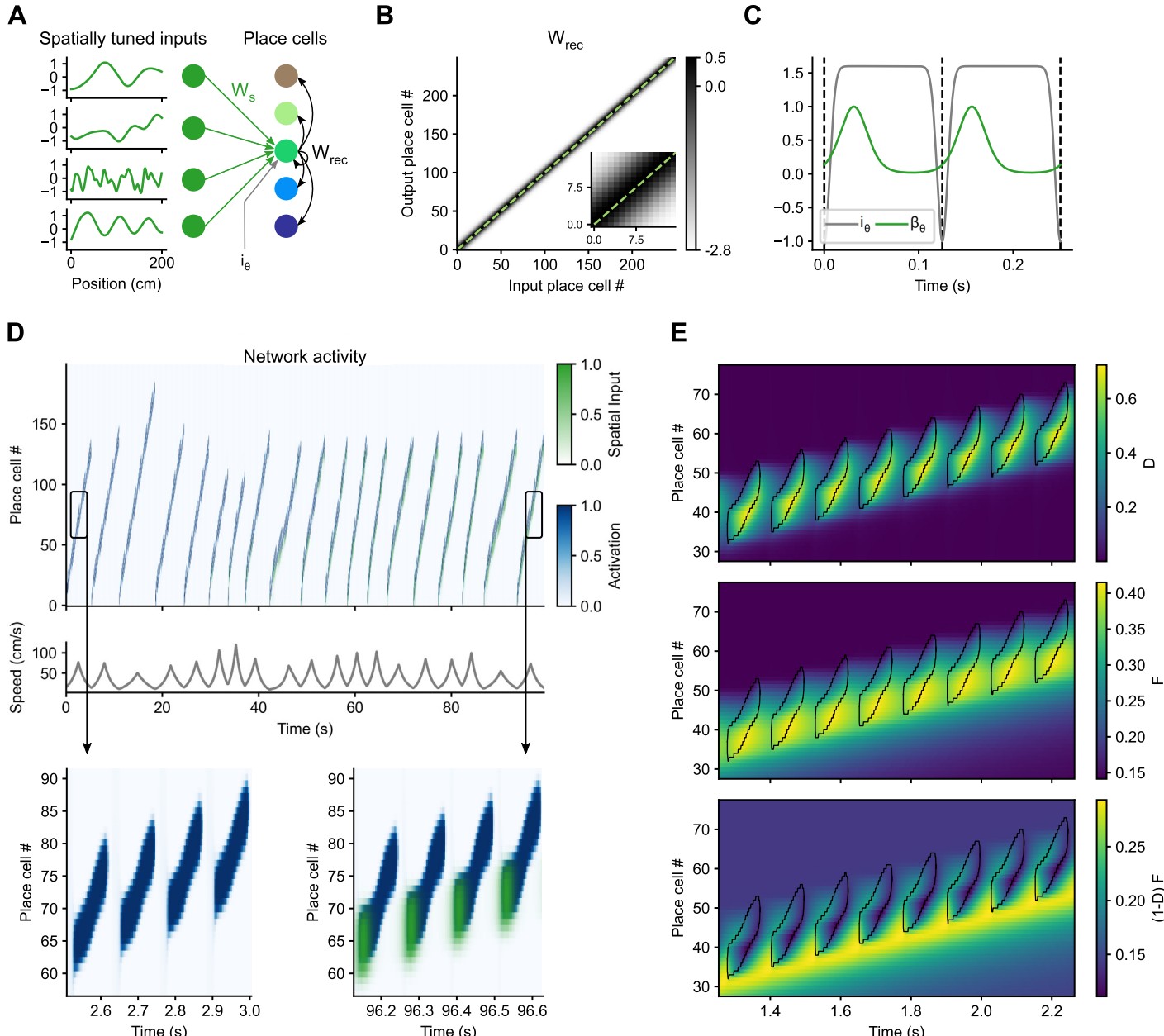

**Fig 3. Elastic mapping of space by internally generated sequences.** A: Model architecture. A preconfigured recurrent network of place cells (colored disks and black arrows) receives inputs from weakly spatially tuned cells through plastic weights $W_s$ (green disks and arrows), as well as rhythmic excitation and inhibition $i_\theta$. Weakly spatially tuned cells and place cells take continuous activation values in the ranges [-1, 1] and [0, 1], respectively. B: Recurrent weights matrix. Dashed line represents identity. C: Temporal evolution of $i_\theta$, the rhythmic excitation and inhibition to all the cells in the network, and $\beta_\theta$, which defines the window in which the cells receive spatial inputs. D: The network activations as the simulated rat runs multiple laps on a novel linear track. The activity in the network is initialized at the first place cells at the beginning of each lap. In the first few laps, activity propagates further in the network in slow laps that take longer to complete. However, in later runs, spatial inputs (green) arriving at the beginning of each theta cycle anchor the activity of the network in space. E: Regions of high activation ($> 0.5$) in the network are contoured in black and overlaid onto the short-term synaptic depression, $D$, and facilitation, $F$, of the outgoing recurrent synapses of the place cells. When emerging from the rhythmic inhibition, an activity bump forms at the cells whose recurrent connections are most facilitated and least depressed, $(1 - D)F$ (bottom panel). This ensures that the each theta sequence starts slightly ahead of where the last one started. Note that due to the threshold used for plotting the black contour, the activity bump actually forms slightly before it, within the yellow band.

formed. This occurs because the recurrent connections between those units are the most effective, since they have accumulated facilitation across several previous theta cycles and have largely recovered from the depression they accumulated in the previous theta cycle (Fig 3E).

We simulated a rat running multiple laps on a 2 m long linear track. Each position along the track has a target mean running speed which follows a triangular profile, being 15 cm/s at both ends and 80 cm/s at the middle. The instantaneous speed of the rat was determined at each time step by multiplying the target mean speed at the current position with a noise term that evolves smoothly over time, thus generating lap to lap variability in speed (Fig 3D, middle).

At the beginning of each lap, activity was initialized at the first neurons in the network and allowed to evolve. The first few laps, activity propagated at a fixed pace in neuronal space, entirely driven by the network's internal dynamics. Thus, activity propagated further in the network the longer it took for the simulated rat to complete a lap (Fig 3D, top left, 0–45 s). At this stage, cells in the network behaved like time cells with time fields anchored to the beginning of the lap; their place fields shifted back and forth depending on running speed. The activity of the cells then became anchored in space by learning the connections between the weakly spatially tuned inputs and the active place cells through a modified Hebbian learning rule. The learning was relatively slow so that it could average out variations in the running speed of the rat at each position. The spatial inputs arrived only within a small temporal window at the beginning of each theta cycle (Fig 3C, green curve), and this is also the window in which the synapses were plastic, consistent with different phases for encoding and retrieval [67, 68]. After learning had taken place, the spatial inputs arriving at the beginning of each theta cycle (Fig 3D, bottom right, green) pulled the activity bump from the position in the network in which it would have formed based only on the network's internal dynamics, towards the position that has become associated with the current location of the animal, thus anchoring the activity of each unit in space. In particular, if the animal ran faster than average, the units in the network associated with the current location of the animal would be further ahead than those that would have become active based only on the network's internal dynamics, and so the spatial inputs would pull the activity bump forward. Conversely, if the animal ran slower than average, the spatial inputs would pull the activity bump backwards. This means that the jump in the initial location of the activity bump from one theta cycle to the next ceased to be constant (Fig 3D, top right, note how the slope of the place cell number vs. time relationship changes for fast vs. slow laps). More precisely, the speed at which the activity bump propagated across theta cycles in neuronal space, $u$, became modulated by the ratio between the instantaneous running speed, $v$, and the average running speed at each position, $\bar{v}$, i.e., $u = \frac{v}{\bar{v}} u_0$. For the remainder of the analyses, we focused on these periods where the learned mapping between physical and neural space had stabilized.

## Effect of speed on place field properties

We first analyzed the properties of place fields produced by the network. To obtain an analogue of firing rate maps, we discretized the linear track and calculated the average activation of each neuron in each spatial bin (Fig 4A). Using these activation maps, we calculated place field sizes and observed that they increased linearly with the mean running speed through the fields (Fig 4B) in agreement with experimentally observed behavior-dependent sweeps [15].

We then examined phase precession by also discretizing theta phase and calculating the average activity of each neuron in each space × phase bin (Fig 4C). Phase precession was more pronounced in the first part of the place field, as observed experimentally [69]. This reflects the fact that the first part of the place field is entirely the result of theta sequences, whereas the last

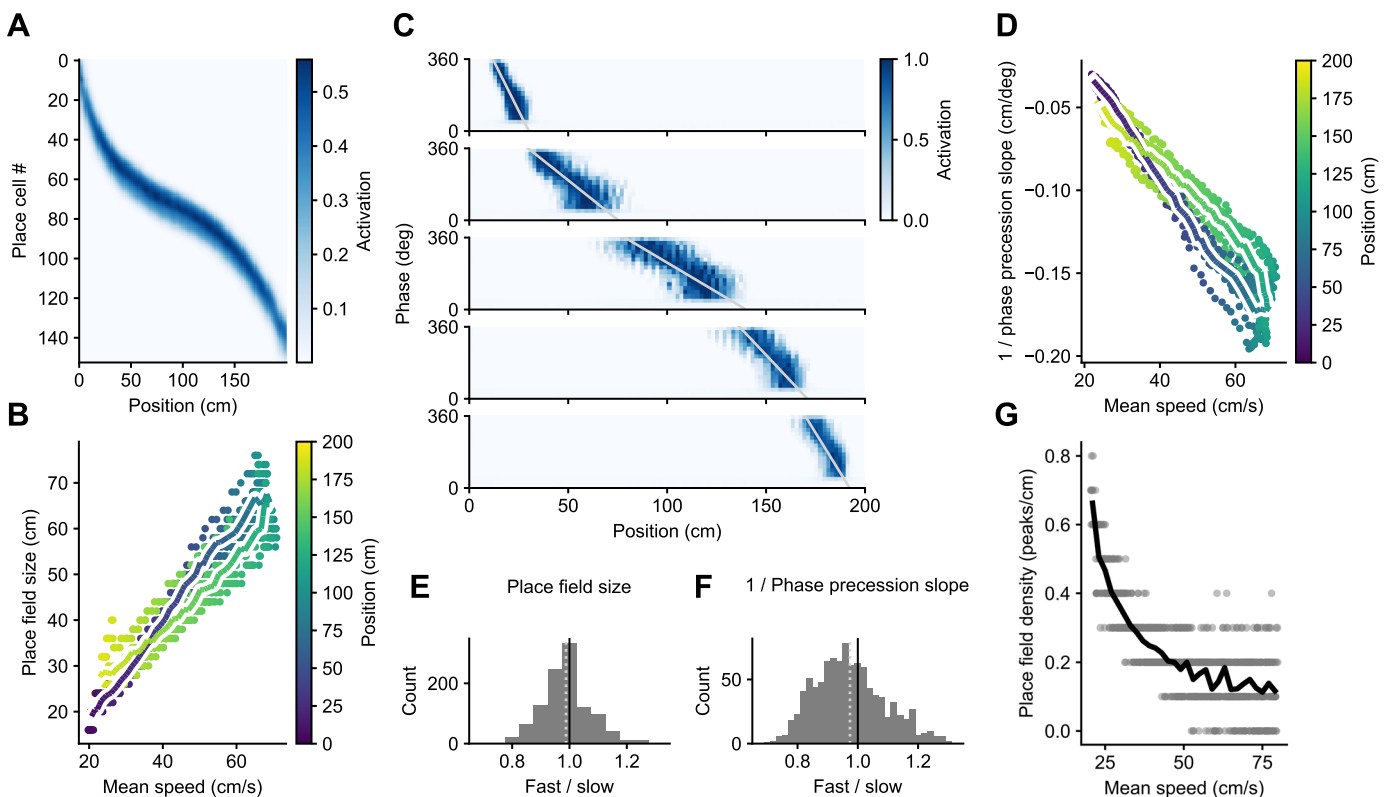

**Fig 4. The model develops larger place fields with shallower phase precession in areas of higher running speed.** A: Mean activation of each unit at each position along the track for one run of the simulation. B: Place field sizes increase as a function of the mean running speed through the place field. Each dot corresponds to a place field, pooled across 10 different simulation runs. The color of the dot represents the position of the place field peak. The coloured line shows the average place field size at each spatial bin along the track with at least 8 samples. C: Examples of theta phase precession from the same simulation run as in A. D: Theta phase precession becomes shallower where the animal runs faster. E, F: Individual place field sizes and inverse phase precession slopes are largely invariant of instantaneous running speeds. Shown is a histogram of ratios of these measures calculated using the top and bottom 20% of instantaneous running speeds at every location. There is a tendency for a modest decrease as the mean of the distribution (dotted line) is below a ratio of 1. G: The density of place fields calculated in 10 cm sliding windows decreases as a function of the mean speed in the window.

part is mostly determined by spatial inputs arriving at the early phases of the theta oscillation. Again in agreement with experimental observations, phase precession was shallower in the middle of the track, where running speed is higher [15]. Since it is the spatial extent of the phase precession clouds that changes linearly with mean running speed, plotting the inverse of the phase precession slope (cm per degree) also changes linearly with mean speed (Fig 4D). Note, however, that for the same mean speed, place field sizes are a bit larger and phase precession a bit shallower in the first half of the linear track, where the animal is accelerating. We comment on this effect in the discussion.

The behavior-dependent sweep model indicates that place field sizes and phase precession slopes do not depend on the instantaneous running speed of the animal on a given lap [15, 70–72]. Thus, we calculated place field sizes and phase precession slopes separately for the top and bottom 20% of speeds in each spatial bin, and plotted their ratios (Fig 4E and 4F). Although the mean of the ratios were slightly below 1, meaning that place fields were smaller and phase precession steeper when animals ran faster, the deviations were small. We therefore conclude that place field properties are largely invariant of instantaneously running speed, suggesting that the current model provides a good approximation to the experimental data.

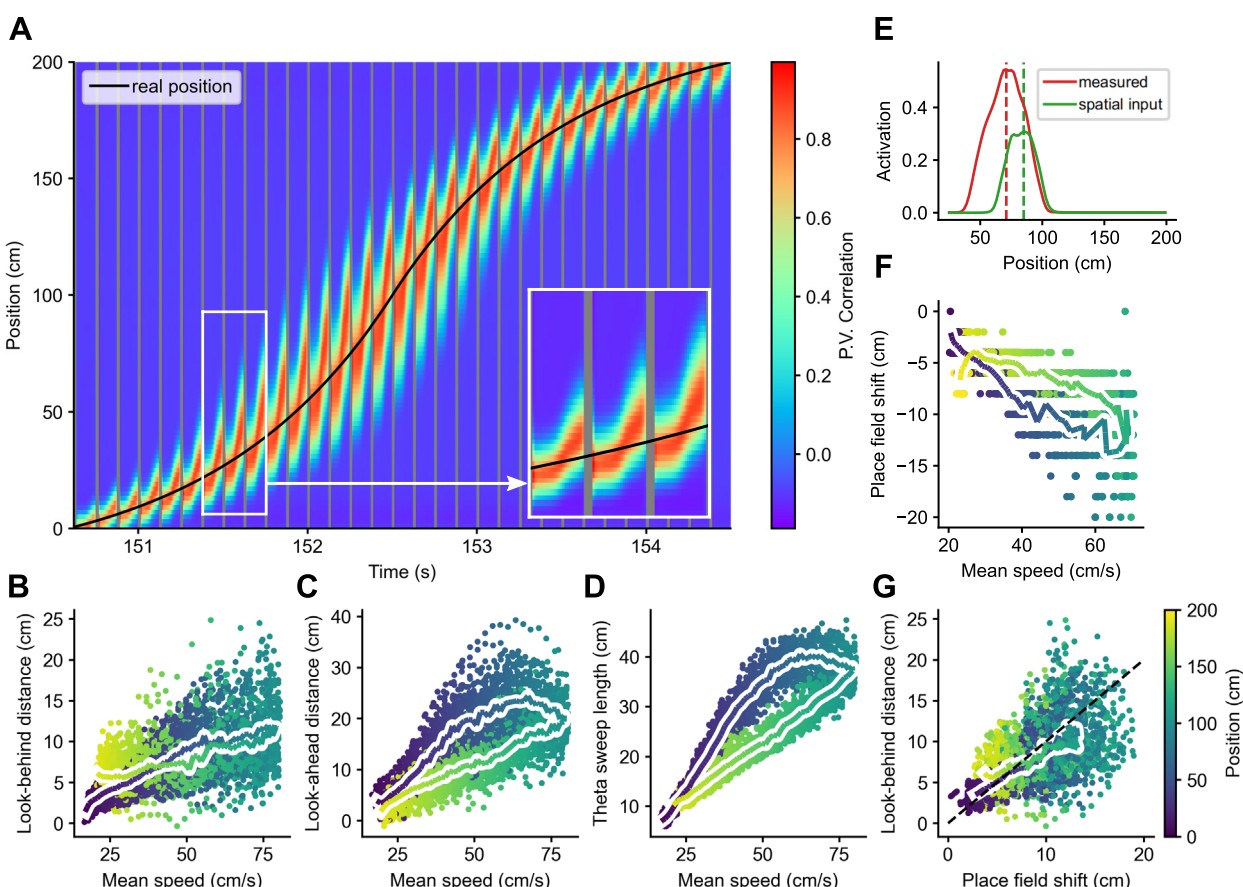

**Fig 5. Decoding analysis.** A: Decoding represented positions from a well trained network during one lap. B: The look-behind distance (how far behind the real position of the animal the decoded sweep starts) increases with the mean speed at the corresponding real position (color coded). Each dot corresponds to a theta sweep, pooled across laps and across 10 different simulation runs. The coloured line shows the average look-behind distance at each spatial bin along the track. C: Same for the look-ahead distance (how far ahead of the real position of the animal the decoded sweep ends), and D: for the total length of the sweep. E: An example of a measured place field that was calculated from the activation of a cell and of a true place field that was calculated based on the spatial inputs that drive the cell. The dashed lines mark the positions of their peaks, and the difference between them is what we call the place field shift. F: Shifts between each place field and its spatial input field. Same conventions as in Fig 4B. G: The look-behind distance of theta sweeps at a given location corresponds roughly to the place field shift at that location. The dashed black line represents identity.

Finally, we examined the effect of mean running speed on place field density by counting the number of place field peaks in a 10 cm sliding window. Consistent with our expectations (Fig 2C), the density of place fields decreased with mean speed following a hyperbolic curve (Fig 4G).

## Effect of speed on theta sweeps

Next, we decoded the position represented by the population of the place cells in our model at each time point. To do this, we correlated the population vector containing the activation of each unit in the to-be-decoded time step with the population vectors containing the mean activation values of the units at each spatial bin along the track [7] (Fig 5A). As commonly observed, the decoded sweeps started behind the current position of the animal and reached ahead of it. We then proceeded to quantify these look-behind and look-ahead distances, as well as the total extent of the sweep. To avoid artifacts introduced by the beginning and end of

the track, we only considered sweeps that did not reach either of them. In agreement with behavior-dependent sweeps, the three measures increased with the mean running speed at the locations at which they occurred (Fig 5B–5D). There was also an effect of acceleration which we discuss below.

### Decoded positions are shifted backwards

That decoded theta sweeps started behind the current position of the animal is in fact surprising considering how the model was set up: spatial inputs arriving at the beginning of each theta cycle pull the activity in the network towards the units associated with the current position, and activity then propagates forward (Fig 3C and 3D). Hence, the network can represent only current and future positions, but not positions in the past.

The backward shift in the decoded positions was the result of decoding using measured place fields that are shifted backwards with respect to the true place fields, which we define as the positions where they are strongly driven by spatial inputs [35] (Fig 5E). Indeed, like the look-behind, the backward shift between measured and true place fields increases with mean running speed (Fig 5F), and both measures roughly match (Fig 5G). We could reveal this shift in our model because both the measured and true place fields are known. We hypothesize that this shift occurs in experimental data as well and discuss the potential implications below.

### Discussion

Previously, we found that theta sweeps are longer, and place fields are larger with shallower phase precession where animals typically run faster, even though individual place fields do not change with instantaneous running speed [15]. These findings were consistent with, and offered a resolution to apparently contradictory observations on the effect of speed on theta sweeps and place field properties [7, 8, 70–72]. Here we developed and studied a mechanistic computational model that reproduces our experimental observations based on pre-configured sequences mapping out space. Furthermore, our modeling results predict that there is a backward shift between the measured place field relative to the true place field of a place cell, and that this artifact accounts for theta trajectories apparently starting at position behind the current location of the animal.

The effect of speed on place field size and density described by our model can also account for some experimental observations for which a link to speed has not been explicitly recognized. A prominent example of this are the findings by Tanni and colleagues [73]. They report that place field width perpendicular (but not parallel) to a wall increases with distance to the wall. At the same time, the density of place fields decreases with the distance to the wall, and these two effects counteract each other such that the proportion of co-active cells is always the same. This is exactly what our model produces under the assumption that animals run more slowly as they approach or leave a wall (Fig 4B and 4G).

Numerous other studies have also observed a higher density of place fields where animals typically run slower, such as in the starting position and turning points of a maze [49, 58–61]. The positions of rewards also tend to be over-represented by place cells [47–57, 60], but see [74, 75]. Some of these studies explicitly report that animals slow down around the rewarded positions, and one of them shows a striking match between the slowdown and the increase in place field density [55] (their Fig 5A–5C).

The smaller and more densely packed place fields that our model produces in areas where animals typically run slower makes intuitive sense for two reasons. First, because these are generally important areas, where animals may benefit from a more fine-grained representation. And second, because since the animal is moving slowly, it would be wasteful to have large

place fields participating in theta sweeps that predict positions far ahead, since those positions would not be reached soon.

In addition to the effect of speed, the model points to an effect of acceleration. Even at the same mean speed, theta sweeps were a bit longer, and place fields a bit larger with shallower phase precession in the first half of the track where the animal accelerated as compared to the second half where the animal decelerated (Figs 4B, 4D & 5D). That is because theta sweeps extend to the positions the animal will reach a fixed time interval in the future, and, other things being equal, those positions will be further away if the animal is accelerating. Intriguingly, Gupta and colleagues [8] report an increase in the theta sweep look-ahead distance with acceleration consistent with our results (Fig 5C), but they also report the opposite effect for the look-behind distance, which we do not observe. More research is required to elucidate the effect of acceleration on place cell firing.

Theta frequency has also been shown to increase modestly with speed (by about 1 or 2 Hz within the range of speeds assessed in experiments; [76–78]), or perhaps with acceleration [79], but see [80]. However, there appear to be compensatory mechanisms in place such that changes in theta frequency do not affect the theta phase code [15, 70–72, 81, 82]. In our model, these compensatory mechanisms would need to ensure that the speed of propagation of the activity bump changes in tandem with theta frequency, perhaps mediated by the increase in firing rate that is also known to occur with speed.

Although this study has focused on the effect of movement dynamics, other factors also appear to influence the hippocampal code. For instance, longer theta sequences have been observed when traveling to positions further away [83], or when correctly retrieving the memorized position of a goal [14]. A higher density of sensory cues may also lead to smaller place fields [84, 85]. Furthermore, it has been proposed that the hippocampus plays a more general role in storing sequences of events extending beyond the spatial domain [37, 86, 87]. Indeed, our model could operate with any kind of input, extracting features that change at a pace that matches that of the network's internal dynamics.

Our model is based on internally generated sequences. Some evidence for this comes from the observation that the hippocampus develops sequences coding for time or distance run in the absence of changing sensory stimulation [40, 41, 88]. Furthermore, the existence of preplay [42–46, 89] and related phenomena [90, 91] points to pre-configured internally generated sequences that are recruited to encode spatial or episodic memories [37, 39]. These views, however, remain controversial [92, 93]. Also, internally generated sequences might not be the only option for modeling behavior-dependent sweeps. One possibility might be to learn the connections between the cells participating in an externally driven sequence (Fig 2B, top) using a relatively long plasticity window, such as in behavioral timescale synaptic plasticity (BTSP) [94, 95]. This would allow for cells coding for positions further apart to connect to each other where the animal runs faster, leading to longer theta sequences and bigger place fields in those areas. Alternatively, instead of asymmetric recurrent connections like in our model, one could use independently phase precessing cells to generate sequential activation of the population [25]. These cells could acquire their spatial selectivity through a BTSP-like learning rule, which would result in larger place fields where animals run faster. Nevertheless, internally generated sequences offer a parsimonious account of both preplay [96] and theta sweeps [36]. Furthermore, the fact that our model, operating on these ideas, reproduces unrelated experimental findings on place field properties offers an indirect line of evidence in favor of internally generated sequences. It is also possible that the cells we have modeled, which participate in internally generated sequences, constitute a backbone onto which more cells are later recruited by other mechanisms [44].

Our model makes some predictions regarding the behavior of place cells in novel environments that could be used to test its validity. Most notably, the model predicts that place cells would express place fields, phase precession and theta sequences from the onset, but the location of the fields would drift initially. Some studies have indeed shown that many place fields are present from the first lap [97, 98], but that firing rate maps are initially unstable [99, 100]. A study has shown that cells show phase precession from the first lap, but only appear to start participating in theta sequences from the second lap [10]. However, in that study, animals ran more slowly during the first lap. According to our model, this would produce steeper phase precession and, correspondingly, short theta sequences, which might be difficult to detect. More work is required to test the model predictions.

Some modeling decisions could also be replaced by other mechanisms or improved to better account for experimental data. For instance, the model uses short-term synaptic facilitation and depression to produce a jump-back in activity at the beginning of each theta cycle. However, similar results could be obtained using other mechanisms such as rebound spiking [101] or slow changes in conductance that produce depolarization in neurons that spiked ca. 100 ms before [16, 102]. A possible extension of the model could be to ensure that place field sizes and phase precession slopes remain completely shielded from changes in instantaneous running speed. This could be achieved by allowing the instantaneous running speed to influence the neuronal propagation of theta sequences (S1 Text), similar to how running speed moves the bump attractor in models of path integration [103, 104]. Finally, the model could be extended to two-dimensional environments. In such a case, different sequences would have to be stitched together to form a coherent representation of space, perhaps similar to how it is done in clone-structured graphs [105].

Previous studies have consistently shown that decoded theta sweeps start behind the current position of the animal [7–15]. Our modeling suggests that this interpretation might need to be reexamined. The issue stems from an inconsistency in the application of decoding techniques to theta sweeps. The commonly used decoders assume that place cell firing always represents the current position of the animal, and so the decoders use the place fields computed from all spikes recorded from each cell to decode the position represented by the population. In contradiction to this assumption, the apparent conclusion of the analysis is that firing at the beginning of the place field, at the late phases of theta, represents a future position, whereas firing at the end of the place field, at the early phases of theta, represents a past position. Therefore, in principle, one should not have used the whole measured place field to decode positions. Ironically, the use of this decoding procedure could still be justified if and only if the conclusion it apparently leads to happens to be correct. This is because if the retrospective and prospective components of the measured place field were of approximately equal size, then the measured and true place fields would be aligned and it would be acceptable to use one in place of the other for decoding. However if, like in our model, place cells only represented present and future positions, then the measured place field would be shifted backwards with respect to true place field, and this would introduce a misleading backwards shift in the decoded positions (Fig 5E–5G). Hence, the observation that theta sweeps start behind the position of the animal might be, at least in part, an artifact of the decoding analysis.

The view that place cells code for present and future positions [33–35] is consistent with experimental observations and suggests new experimental tests. Bi- or omni-directional place fields have been observed to shift backwards with respect to the direction of travel [106, 107], and place fields become more compact if this backward shift is compensated for by shifting the position of the spikes forward in space [107] or time [108, 109]. This has generally been construed to show that place cells represent positions slightly ahead of the animal (e.g., positions ca. 120 ms in the future in the study by Muller and Kubie). However, in the context of theta

phase coding, these findings support the view that theta sweeps start at the current position of the animal and reach forward. In such a case, firing at the beginning of the measured place field would correspond to a prediction that the animal will reach the cell's true place field, which would be located towards the end of the measured place field (Fig 5E). Note that the peak or the center of mass of the measured place field would lead the true place field by a relatively small amount (e.g., the 120 ms mentioned above), but the beginning of the place field would correspond to the representation of positions substantially more ahead than that. For example, in our model, the shift between the measured and true place fields is ca. 160 ms, but the beginning of the place field corresponds to the representation of positions ca. 600 ms ahead.

Finally, while it is challenging in practice, in principle it might be possible to examine the relationship between measured and true place fields directly in experiments. The key would be to dissociate recurrent inputs to place cells from external feedforward inputs, whose spatial distribution would correspond to the cells' true place fields. It may also be possible to estimate the locations where spatial inputs drive place cell activity based on the idea that place cells should fire most reliably, i.e., with the lowest variability, in these locations. That is because the external inputs at these locations act as anchor points for the activity of the cells, whereas firing outside of these locations is mediated by internal mechanisms that would presumably introduce some noise in the relationship between position and firing rate. Thus, one could try to find the portion of the measured place field (defined in terms of position and/or theta phases at which the spikes are emitted) that has the lowest trial-to-trial variability in firing rates. Decoding population activity using these portions of place fields should then result in the highest decoding precision, e.g., highest probability values in a Bayesian decoder, as well as accuracy.

In conclusion, we have developed a model of behavior-dependent sweeps in the hippocampus that is consistent with the observation of preplay and that accounts for the effect of speed on theta sweep lengths, place field sizes, phase precession slopes and densities. Furthermore, the modeling suggests that theta phase coding might be limited to current and future positions, causing place fields measured from hippocampal recordings to systematically misrepresent the true nature of spatial responses of hippocampal place cells.

## Materials and methods

### Linear track and simulated motion

We simulated a rat running 30 laps on a novel 200-cm-long linear track using 1 ms time steps. When the rat reaches the end of the track, it is teleported back to the beginning. The target mean running speed of the rat at each position along the track has a triangular profile, being 15 cm/s at both ends, and 80 cm/s in the middle of the track. To generate variability in the running speed, each time step we multiply the target mean speed at the current position of the rat with a noise factor that evolves slowly. This noise factor is obtained by convolving independent samples of Gaussian noise (mean = 0, std = 1) at every time step with a Gaussian kernel (mean = 0 s, std = 2 s), and scaling and shifting the result so that it ranges between 0.5 and 1.5 and is centered at 1.

We then create 128 weakly spatially tuned features, $s$, defined over the extension of the track, discretized in 1 cm bins. These features are produced by filtering independent Gaussian noise (mean = 0, std = 1) samples in each spatial bin with Gaussian kernels (mean = 0, std randomly selected between 2 and 20 cm). Each of the signals is then scaled and shifted so that it ranges between -1 and 1 and is centered at 0.

## Network

We simulated a network composed of 250 units, whose activation level, $r$, follows the differential equation:

$$\tau_r \frac{dr}{dt} = -r + i_\theta + W_{\text{rec}}((1-d) \odot f \odot \sigma_r(r)) + i_{\text{init}} + i_{\text{ext}}, \tag{1}$$

where $\tau_r = 5$ ms. $i_\theta$ provides rhythmic excitation and inhibition to all units in the network as a function of the phase $\theta$ of the 8-Hz theta oscillation. Excitation peaks around $\theta = \pi$, providing a temporal window where activity bumps can form and propagate; inhibition is maximal around $\theta = 0$, shutting down all activity in the network. $i_\theta$ is illustrated in Fig 3C, and given by the following equation:

$$i_\theta = -(i_\theta^{\text{max}} - i_\theta^{\text{min}})e^{\kappa_\theta \cos(\theta)}/e^{\kappa_\theta} + i_\theta^{\text{max}}, \tag{2}$$

where $\kappa_\theta = 12$, $i_\theta^{\text{max}} = 1.6$, and $i_\theta^{\text{min}} = -1$. The third term on the right hand side of Eq 1 defines the recurrent interactions between units in the network. These are the result of multiplying the recurrent weights matrix, $W_{\text{rec}}$, with the activity of the units after applying a logistic activation function, $\sigma_r(r)$, and multiplying element-wise by each unit's short-term presynaptic facilitation, $f$, and the complement of the depression, $1 - d$. The recurrent weights from unit $j$ to unit $i$ are given by:

$$W_{\text{rec},ij} = (w_{\text{rec}}^{\text{max}} - w_{\text{rec}}^{\text{min}})e^{-\frac{(i-j-\delta)^2}{2\sigma_{\text{rec}}^2}} + w_{\text{rec}}^{\text{min}}, \tag{3}$$

where $\delta = 0.6$ controls the direction and speed of propagation of activity bumps in the network, $\sigma_{\text{rec}} = 5$, $w_{\text{rec}}^{\text{max}} = 0.5$, and $w_{\text{rec}}^{\text{min}} = -2.8$. The logistic activation function is given by:

$$\sigma_r(x) = \frac{1}{1 + e^{-\alpha_r(x-x_r)}}, \tag{4}$$

where $\alpha_r = 6$, and $x_r = 0.5$. The short-term presynaptic facilitation and depression follow:

$$\tau_f \frac{df}{dt} = -f + (1-f)\sigma_r(r) + f_0 \tag{5}$$

$$\tau_d \frac{dd}{dt} = -d + \sigma_r(r), \tag{6}$$

where $\tau_f = 0.32$ s, $\tau_d = 0.06$ s, and $f_0 = 0.14$.

At the beginning of each simulated lap, $r$ and $d$ are set to 0, and $f$ is set to $f_0$. $i_{\text{init}}$ is then used to initialize activity in the network. $i_{\text{init}} = \beta_\theta$ for the first five units in the network for the duration of one theta cycle, and otherwise equal to 0. $\beta_\theta$ is illustrated in Fig 3C and described by the following equation. It takes on values between 0 and 1 across the theta cycle, peaking at $\theta = \pi/2$:

$$\beta_\theta = e^{\kappa_s \sin(\theta)}/e^{\kappa_s}, \tag{7}$$

where $\kappa_s = 2$.

The last term in Eq 1, $i_{\text{ext}}$, corresponds to the spatial input resulting from the matrix multiplication of the plastic weights, $W_s$, and the weakly spatially tuned features described in the previous section, $s$, after applying a 'dendritic' nonlinearity, $\sigma_s$:

$$i_{\text{ext}} = \beta_{\text{max}}\beta_\theta\sigma_s(W_s s) \tag{8}$$

where $\beta_{max} = 0.4$. The dendritic nonlinearity is given by the logistic function Eq 4 with parameters $\alpha_s = 12$ and $x_s = 0.7$.

The weights, $W_s$, are initialized to 0 and updated based on a modified Hebbian plasticity rule:

$$\tau_w \frac{dW_s}{dt} = \beta_\theta s \otimes (\sigma_r(r) - i_{ext}),\tag{9}$$

where $\tau_w = 20$ s. The weight change is gated by $\beta_\theta$, and depends on the outer product of the spatially tuned inputs and the activations of the units minus $i_{ext}$. The subtraction of $i_{ext}$ acts as a normalizing factor that limits the growth of the weights to the value required to match the bounded activity of the units.

## Analysis of place field properties

For the following analyses, we discarded the first 80 s of the simulation, where place fields are unstable. For calculating the units' measured place fields, we discretized the track in 2 cm bins and computed the mean activation $\sigma_r(r)$ of each unit in each bin. The mean activations are then smoothed slightly by convolving them with a Gaussian kernel (std = 3 cm). We discarded units with a peak activation below 0.2, and units with place fields substantially cut off by the beginning or end of the track, i.e., place fields for which the activation value did not go below 50% of the peak value on both sides.

We calculated place field sizes as the extent of the place field above 10% of the peak value. For place fields where this threshold was only reached on one side of the peak, we calculated place field size as twice the distance from the peak to that point [15]. For the rest of the analyses, we only included place fields that did go below the 10% threshold on both sides of the peak.

We computed true place fields in the same way, but using the modulation of the inputs to the place cells, $i_{ext}$, instead of their output $\sigma_r(r)$. Then we computed the place field shift as the difference between the peaks of the measured and true fields.

To analyze phase precession, we discretized the track using 2 cm steps and the theta phase using 20° steps, and computed the mean activation of each unit in each position × phase bin. We computed the slope of phase precession by fitting lines to the phase precession clouds using orthogonal distance regression as implemented in the SciPy library [110]. This algorithm minimizes the orthogonal distance of the points to the best fitting line. Because of this, it can accurately describe very steep phase precession clouds that would result in approximately horizontal fits when using other linear regression methods that only minimize the deviations between the points and the best fitting line along the $y$ axis [15].

We computed the change in individual place field sizes and phase precession slopes with instantaneous speed by independently calculating these values using only time steps with instantaneous speeds in the top or bottom 20% of speeds at each spatial bin. For this analysis, we increased the number of laps to 45.

For calculating the density of place fields, we counted the number of place field peaks in overlapping 10 cm windows with a stride of 2 cm.

## Decoding

We decoded positions from the network activity at each time step by calculating, for each spatial bin, the match between the instantaneous population activation and the mean activation of the units at that spatial bin, as assessed through the Pearson correlation coefficient [7]. The spatial bin with the highest correlation value was selected as the decoded position at that time

step. We only decoded position in time steps where at least one unit had an activation value above 0.1. This excluded periods between theta cycles when the units in the network were strongly inhibited.

### Analysis of theta sweeps

To avoid artifacts introduced by the boundaries of the environment, we excluded theta cycles in which either the first or the last spatial bin achieved the highest correlation value. For each theta cycle, we identified the time steps corresponding to the first and last 36° bin with valid decoding values, i.e., the beginning and the end of the theta sweep. The look-behind distance was then defined as the difference between the mean real position and the mean decoded position within the first temporal window. Similarly, the look-ahead distance was defined as the difference between the mean decoded position and the real position within the second temporal window [8]. The overall theta sweep length was defined as the difference between the mean decoded positions in the second and first temporal windows.

## Supporting information

**S1 Text. Making place field sizes invariant of instantaneous speed.** This supplementary text contains a mathematical derivation showing how place field sizes and phase precession slopes can remain constant despite changes in instantaneous running speed if the speed of propagation of the activity bump within theta cycles is made dependent on instantaneous running speed.
(PDF)

## Author Contributions

**Conceptualization:** Eloy Parra-Barrero, Sen Cheng.

**Formal analysis:** Eloy Parra-Barrero.

**Funding acquisition:** Sen Cheng.

**Methodology:** Eloy Parra-Barrero, Sen Cheng.

**Project administration:** Sen Cheng.

**Software:** Eloy Parra-Barrero.

**Supervision:** Sen Cheng.

**Visualization:** Eloy Parra-Barrero.

**Writing – original draft:** Eloy Parra-Barrero, Sen Cheng.

**Writing – review & editing:** Eloy Parra-Barrero, Sen Cheng.

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
