## [Decision Letter · Decision Letter 0]

14 Feb 2023

Dear Dr. Cheng,

Thank you very much for submitting your manuscript "Learning to predict future locations with internally generated theta sequences" for consideration at PLOS Computational Biology. As with all papers reviewed by the journal, your manuscript was reviewed by members of the editorial board and by several independent reviewers. The reviewers appreciated the attention to an important topic. Based on the reviews, we are likely to accept this manuscript for publication, providing that you modify the manuscript according to the review recommendations. In particular, there are several aspects of the model and results that need to be described more clearly, in line with the reviewers concerns. 

Sincerely,

Daniel Bush

Academic Editor

PLOS Computational Biology

Lyle Graham

Section Editor

PLOS Computational Biology

Reviewer's Responses to Questions

**Comments to the Authors:**

Reviewer #1: The authors proposed a computational model for explaining their previous findings published in eLife, where they showed that as the animal runs faster, place fields are wider and phase precession is shallower in those areas. Specifically, the model consists of a continuous attractor network with short-term synaptic facilitation and depression that internally generates theta sequences that advance at a fixed pace. Spatial locations are then mapped onto the active units via modified Hebbian plasticity, so that neighboring units end up representing spatial locations further apart where animals run faster, reproducing their earlier experimental results. The model also explains over-representation of locations when animals slow down in those areas. The novel contribution of this paper is a computational model of position mapping through Hebbian learning, which provides a way of hippocampal information coding that arises from the interplay between sensory input and predefined network dynamics.

Below are the major/minor points I would love the authors to answer.

Major points:

1, The function of different parts of the model is not well introduced. In the previous Tsodyks’ model, the activity bump is driven by the external moving input, which travels faster in individual theta cycles either due to the asymmetric connection (Tsodyks et al, 1996) or STD (Romani and Tsodyks, 2015). However, the authors pointed out that the activity bump reappeared slightly ahead of where the last one formed. Can the authors explain the role of the external input in this model and if the bump's location is driven by the external input or by internal mechanisms such as STD/STF? Additionally, can the authors provide information on how the model's hyper-parameters were tuned to match the intrinsic speed of the bump with the external speed of the artificial animal (if it is the case)?

2, The authors mention that Hebbian learning shapes the density of spatial coding of individual neurons, but do not specify how this affects the speed of the activity bump in the neuronal space. Can the authors explain if the activity bump always travels with a constant speed in neuronal space regardless of the animal's running speed?

3, Theta frequency is known to vary with running speed. However, in the continuous attractor, the rhythmic input is set as a fixed frequency. Can the authors explain how this may affect the phase precession slope and if the model's conclusions still hold in cases where theta frequency is varied with running speed.

4, I am confused by the instantaneous running speed in their analysis (specifically in Fig. 4E&F). Can the authors explain what is meant by instantaneous speed and how it differs from the non-instantaneous speed (or the characteristic speed mentioned in the authors' previous eLife paper)? Additionally, can the authors provide insight on how their conclusions would change if the analysis was done using for example, the top and bottom 20% of running speeds instead of the top and bottom half of instantaneous speeds (in the plot of Maurer et al. (2012) (Fig. S1), they used low (20-40cm/s) and high (80-100cm/s) velocity bins to show place field size is invariant against running speed)?

Is there any experimental evidence supporting the internal structure of place cells in CA3 as a continuous attractor network (CAN)? The preplay papers only indicate internally generated sequences in the HPC, not a CAN.

Minor points:

1, The title is confusing. The network model with STD/asymmetric connections (Tsodyks’ model) can internally generate bump sweeps, which does not require any learning. Moreover, there seems no prediction of future locations, since it is a 1D linear track environment (see Johnson and Redish, 2007, JNS). Can the authors provide a clearer title that reflects the novel contribution of the paper?

2, Fig. 2B is not clear. Better to split into two plots with one showing the internally driven sequence and the other showing the externally driven sequence.

3, Fig. 2B is not clear. It would be beneficial to split the figure into two plots, one showing the internally driven sequence and the other showing the externally driven sequence, to improve clarity.

4, Why is the spatial input also rhythmic? What if it is a constant input with changing location?

5, Could the change of the place field size with running speed be an analysis artifact? The usual way of calculating the place field size is to calculate the firing rate at each spatial bin and then cut a threshold to determine the firing field. When animals run slower, the firing rate is lower, then cutting the same threshold as when the animals run faster will lead to a smaller place field.

Overall, the computational model is promising. However, the results appear to be a direct replication of previous work and may require further verification. My recommendation is for a major revision.

I hope that the authors will find my feedback helpful in improving their work.

Reviewer #2: This study focuses on the importance of representing past, present, and future locations in spatial navigation. The researchers observed that populations of place cells in the hippocampus tend to show trajectories starting from behind the animal's current location, moving ahead during each cycle of the theta oscillation. The findings suggest that the position represented by the CA1 place cells during a specific theta phase corresponds to where the animal was or will be located at a fixed time interval in the past or future, provided the animal is moving at its normal speed and not its current speed. This behavior-based phenomenon results in longer theta trajectories, bigger place fields, and shallower phase precession in areas where the animal typically moves faster.

The researchers have created a computational model to explain these observations. The model is a continuous attractor network that generates theta sequences at a constant pace through short-term synaptic facilitation and depression. Spatial locations are assigned to the active units through modified Hebbian plasticity, causing neighboring units to represent farther apart locations where the animal moves faster. The model also accounts for the higher density of place fields typically seen when the animal slows down, such as near rewards.

The results of the model suggest that an artifact of the decoding analysis may contribute to the observation that theta trajectories start from behind the animal's current position. In conclusion, the study provides deeper insight into how the hippocampal code arises from the interaction of behavior, sensory input, and inherent network dynamics.

Broadly, I have no concerns about the research itself. However, there is some pertinent information that the authors should address prior to publishing their manuscript to ensure they are discussing data that both agree and disagree with their model.

1. The authors make much of the argument that prior research has suggested that sweeps start from behind the animal and proceed to locations ahead of them. If one were to ask a bipedal animal where they are in Cartesian coordinates with respect to the floor, they would most-likely point to their feet. In a quadrupedal animal, it is inherently more complicated to ask the animal “where exactly are you?” Do they consider their position to be their head (where that investigator always decodes the position – resulting in sweeps initiating from behind) or the other extreme, their tail (where all sweeps would be prospective)? Perhaps something in between? Without explicitly knowing where the animal considers to be their “absolute location,” the emphasis on where the sweep initiates seem forced and artificial.

2. There is reason to believe that, when animals firsts enter an environment, all known speed modulations are absent (Environmental novelty is signaled by reduction of the hippocampal theta frequency; Jeewajee et al., 2007). This is potentially problematic as it would suggest that the physical space mapping in figure 1C could be uniformly distributed in a manner that matches the neuronal space (rather than clustered at the end of the tracks). Otherwise, the authors would need to account for why the uneven distribution occurs a priori to when the rat knows its going to run fast or slow at a certain location.

3. In a recent paper, the author’s assumption that speed modulates theta sequences is directly challenged (Hippocampal place cell sequences differ during correct and error trials in a spatial memory task; Zheng et al., 2021). Zheng and colleagues state that as the running speed at the stopping point was the same in both successful and unsuccessful trials, differences in running speeds were not the cause of the differences in sequence slopes between successful and unsuccessful trials. How do the authors reconcile these results with their current manuscript? Certainly, addressing the Zheng results in the current manuscript seems pertinent.

4. While it is widely accepted that speed (with the manuscript described in point 3 above being among the outliers) is the factor responsible for setting the size of place fields/slope of precession/ theta sequences (effectively, all being different measures of the same phenomenon), placing objects has been found to result in a dramatic change in both place field size and phase precession slope (The Influence of Objects on Place Field Expression and Size in Distal Hippocampal CA1; Burke et al., 2011). These data also challenge the idea that speed is the only pertinent factor. Moreover, one could use these results to explain the results in figure 5. Leaving a goal location, the animal is quite confident about their location as they are departing a food reward. They begin to traverse a cue poor region of the track, resulting in higher uncertainty/larger sweeps. However, as they approach the goal, they can see the end of the track, and the reward and fields get small as the rat decreases the speed and approaches the next reward.

5. The above explanation seems to be as tenable as the discussion of acceleration. The discussion on acceleration seems almost like an afterthought, with no citations. While there is data from Kropff et al. that offers it is the driving factor of theta frequency, running counter to the mass of prior literature (Frequency of theta rhythm is controlled by acceleration, but not speed, in running rats; 2021), these data have been called into question by Kennedy et al. (A Direct Comparison of Theta Power and Frequency to Speed and Acceleration; 2022). The authors seem to have two pathways in addressing this – either explicating the issue, wading into the conflict, or not mentioning acceleration as it doesn’t seem directly relevant. I would suggest the latter.

6. Finally, I wish for the authors to consider a near-heretical idea. What If the hippocampal correlate is not space per se. And by extension, what If the velocity correlates are related to some other function that the brain is performing? For instance, Buzsaki and Llinas (2017) offer a unique perspective. That while it is easy to derive correlates between the brain's spatial-temporal activity and distance and duration measurements, these observations may be coincidental. They offer that it is not possible for the brain itself to directly sense space or time, concluding that, instead of searching for representations of space and time in the brain based on prior assumptions, it is important to investigate the brain mechanisms that lead to the formation of inferential, explanatory models. This perspective has been extended by Nadel and Maurer (2021), stepping away from space and time, to describe place cells as containers that serve to maintain stimulus equivalence or cognitive continuity from one moment to the next. From these perspectives, the authors observations could be reinterpreted under the assumption that the larger sweeps are related to regions without much change in space/time (velocity being the measure that relates these two things; e.g., cm/s) while smaller sweeps are instances where stimuli change relatively faster.

Reviewer #3: Parra-Barrero and Cheng present a model for theta sequences in which hippocampal neurons internally produce sequences driven by synaptic adaptation and a pre-defined connectivity. Through the learning of associations with spatial inputs, these neurons become place cells and their sequences become theta sequences. The authors find that this model for theta sequences reproduces experimental observations in that theta sweeps are longer where the animal typically moves faster and place cells are more numerous where the animal typically moves slower.

Their underlying model represents a meaningful contribution towards the search for the mechanisms that produce theta sequences. It provides an explanation for experimental data and presents a useful theoretical construct. Its conclusions are presented and justified well. However, addressing two major points, at least by discussion if not by new analyses, would improve its significance and its connection to the literature.

Major point 1: One claim is that reports of theta sequences starting from past positions are artifacts of a naive decoder and that the true sequences should start from the current animal position. This is a substantial statement. While this possibility has been identified, it would be helpful if a solution were proposed. In particular, in the authors' own simulated data, these sweeps also start from past positions. Could they create a decoding scheme such that these sequences would start from the current position? Ideally, this would not involve knowledge of spatial input, whose experimental measurement is often intractable. Would a decoder incorporating the theta phase of activity or the animal's speed be helpful?

Major point 2: A worthwhile feature of this model is the implementation of learning over experience in a new environment (Fig. 3). Are there predictions concerning phase precession or theta sequences in this regard that have been or can be tested by experiments? Or is the learning process idiosyncratic to the model and is only meant to produce the appropriate learned network? The authors hinted at some features in their model, but a bit more thorough treatment would be beneficial. Their proposal that learning occurs in the associations with spatial inputs, and not the connections between place cells, is a major thesis of this work, and it seems natural that the learning mechanism in the model can be best tested through the analysis of learning in experiments.

Minor point 1: It is a bit difficult to see that activity bumps form at the location of highest (1-D)F in Fig. 3E. There is a band of high value that seems to lie right behind the formation location. Further clarification would be helpful.

Minor point 2: It would be helpful to include an explicit definition of "true place field" in Section 2.4. Although this term has been used by Sanders et al. (2015), it is not universal and can sound a bit overreaching.

Minor point 3: I could understand the Supplemental Material, but it would be helpful to motivate the calculation of theta sequence speed in network units earlier in the text. Also, it would be helpful to use different variable name in Eq. 11 or 14, since $s$ is used for different quantities (albeit with different subscripts).

**Have the authors made all data and (if applicable) computational code underlying the findings in their manuscript fully available?**

Reviewer #1: Yes

Reviewer #2: Yes

Reviewer #3: **No: **

PLOS authors have the option to publish the peer review history of their article (what does this mean?). If published, this will include your full peer review and any attached files.

Reviewer #1: **Yes: **Zilong Ji

Reviewer #2: No

Reviewer #3: No

Figure Files:

Data Requirements:

Reproducibility:

References:

---

## [Decision Letter · Decision Letter 1]

13 Apr 2023

Dear Dr. Cheng,

We are pleased to inform you that your manuscript 'Learning to predict future locations with internally generated theta sequences' has been provisionally accepted for publication in PLOS Computational Biology.

Best regards,

Daniel Bush

Academic Editor

PLOS Computational Biology

Lyle Graham

Section Editor

PLOS Computational Biology

Reviewer's Responses to Questions

**Comments to the Authors:**

Reviewer #1: I have carefully reviewed the changes in the revised paper and I recommend to publish this paper in PCB. I appreciate the significant effort the authors have put into addressing the concerns raised in my previous review. The revisions have significantly improved the clarity, organization, and overall quality of the manuscript.

Looking forward to see the final version of your manuscript.

Reviewer #2: All of my concerns have been addressed.

Reviewer #3: The authors have convincingly addressed my concerns.

One note is that my Minor point 3 may have been misunderstood perhaps due to ambiguity in my writing. I intended "earlier in the text" to mean earlier in the Supplemental Material, not the main text. In fact, I think the original version of the main text without the introduction of $u$ and $v$ is clearer. Minor point 3 was intended to suggest introducing the goal of calculating the speed of theta sweeps in neural space earlier in Supplementary Material. The title of the section is "Making place field sizes invariant of instantaneous speed," and the mathematical formulation of that statement is a constant $u_\\theta(t)$. I think it would be helpful to draw this connection at the beginning of the Supplementary Material and to state that the following calculations intend to derive $u_\\theta(t)$. I wasn't so sure at the start of the section about the purpose of each step and how it would ultimately connect to place field size.

**Have the authors made all data and (if applicable) computational code underlying the findings in their manuscript fully available?**

Reviewer #1: Yes

Reviewer #2: None

Reviewer #3: Yes

PLOS authors have the option to publish the peer review history of their article (what does this mean?). If published, this will include your full peer review and any attached files.

Reviewer #1: No

Reviewer #2: No

Reviewer #3: **Yes: **Louis Kang

---

## [Editor Report · Acceptance letter]

8 May 2023

PCOMPBIOL-D-22-01848R1 

Learning to predict future locations with internally generated theta sequences

Dear Dr Cheng,

I am pleased to inform you that your manuscript has been formally accepted for publication in PLOS Computational Biology. Your manuscript is now with our production department and you will be notified of the publication date in due course.

With kind regards,

Zsofia Freund
